# Succinate Is an Inflammation-Induced Immunoregulatory Metabolite in Macrophages

**DOI:** 10.3390/metabo10090372

**Published:** 2020-09-15

**Authors:** Karl J. Harber, Kyra E. de Goede, Sanne G. S. Verberk, Elisa Meinster, Helga E. de Vries, Michel van Weeghel, Menno P. J. de Winther, Jan Van den Bossche

**Affiliations:** 1Department of Molecular Cell Biology and Immunology, Amsterdam UMC, Vrije Universiteit Amsterdam, 1081 HV Amsterdam, The Netherlands; k.harber@amsterdamumc.nl (K.J.H.); k.degoede@amsterdamumc.nl (K.E.d.G.); s.verberk@amsterdamumc.nl (S.G.S.V.); e.meinster@amsterdamumc.nl (E.M.); he.devries@amsterdamumc.nl (H.E.d.V.); 2Department of Medical Biochemistry, Amsterdam UMC, University of Amsterdam, Experimental Vascular Biology, 1105 AZ Amsterdam, The Netherlands; m.dewinther@amsterdamumc.nl; 3Laboratory Genetic Metabolic Diseases, Amsterdam UMC, University of Amsterdam, 1105 AZ Amsterdam, The Netherlands; m.vanweeghel@amsterdamumc.nl

**Keywords:** succinate, SUCNR1, immunometabolite, macrophage, inflammation, metabolite

## Abstract

Immunometabolism revealed the crucial role of cellular metabolism in controlling immune cell phenotype and functions. Macrophages, key immune cells that support progression of numerous inflammatory diseases, have been well described as undergoing vast metabolic rewiring upon activation. The immunometabolite succinate particularly gained a lot of attention and emerged as a crucial regulator of macrophage responses and inflammation. Succinate was originally described as a metabolite that supports inflammation via distinct routes. Recently, studies have indicated that succinate and its receptor SUCNR1 can suppress immune responses as well. These apparent contradictory effects might be due to specific experimental settings and particularly the use of distinct succinate forms. We therefore compared the phenotypic and functional effects of distinct succinate forms and receptor mouse models that were previously used for studying succinate immunomodulation. Here, we show that succinate can suppress secretion of inflammatory mediators IL-6, tumor necrosis factor (TNF) and nitric oxide (NO), as well as inhibit *Il1b* mRNA expression of inflammatory macrophages in a SUCNR1-independent manner. We also observed that macrophage SUCNR1 deficiency led to an enhanced inflammatory response without addition of exogenous succinate. While our study does not reveal new mechanistic insights into how succinate elicits different inflammatory responses, it does indicate that the inflammatory effects of succinate and its receptor SUCNR1 in macrophages are clearly context dependent.

## 1. Introduction

Macrophages play a significant role in innate immunity, driving inflammatory responses in health and disease. Upon stimulation of pattern recognition receptors by ligands such as lipopolysaccharide (LPS), macrophages undergo vast metabolic rewiring in order to compensate for changes in energy requirements [1,2,3,4,5]. Such a mechanism, comparable to the Warburg Effect observed in tumor cells, results in a major shift from high oxidative phosphorylation (OXPHOS) to a more glycolytic phenotype, termed aerobic glycolysis [6,7,8]. Krebs cycle intermediates such as succinate, itaconate, fumarate and citrate have all been described to accumulate in inflammatory macrophages [9,10,11]. Interleukin-4 (IL-4)-induced alternatively activated macrophages have been shown to accumulate the metabolite α-ketoglutarate. These accumulated metabolites have the potential to regulate immune responses and could play a role in disease progression. With the new concept of immunometabolism, we now believe there is potential to harness this switch in macrophage metabolic profiles to revert them to a more protective state as a treatment for disease, particularly in inflammatory diseases [12,13,14].

The Krebs cycle metabolite succinate is one of the most well-described immunometabolites in macrophages. Succinate is synthesized within the mitochondrial matrix and is the ligand for electron transport chain (ETC) complex II, supplying ATP synthase electrons to drive ETC through oxidation to fumarate. In spite of this, upon Toll-like receptor 4 (TLR4) activation of macrophages, the canonical Krebs cycle becomes truncated, causing lower SDH activity at the site of ETC complex II. Consequently, succinate oxidation is limited and subsequent accumulation of succinate occurs [11,15]. As a result, mitochondrial ROS production is increased and subsequent stabilization of hypoxic inducible factor 1-alpha (HIF-1α) occurs, the driver of further pro-inflammatory responses in macrophages [16]. Traditionally, HIF-1α is regulated by prolyl hydroxylases (PHDs), where hydroxylation induces targeting of HIF-1α for proteosomal degradation [17]. However, when succinate levels rise, PHDs are inhibited by succinate through product inhibition, activating the HIF-1 transcriptional pathway, which has been described to induce interleukin-1β (IL-1β) secretion in inflammatory macrophages and potentially lead to the development of inflammatory diseases [16]. Although reducing PHD inhibition could alleviate disease progression, there is an alternative pathway by which succinate is able to induce inflammatory responses.

The succinate receptor SUCNR1 is present on the cell surface of many tissue cell types, including kidney, spleen and small intestine cells, while also being expressed in myeloid cells such as dendritic cells (DCs) and macrophages [18]. The SUCNR1 receptor is a G protein-coupled receptor (GPR91) which consists of both a tightly bound βγ dimer and an α subunit [19]. Once succinate binds SUCNR1, the α and βγ subunits dissociate and induce signaling down the mitogen-activated protein kinase (MAPK) pathway, resulting in transcription of a set of genes, differing based on cell type [20]. In myeloid cells, the succinate–SUCNR1 axis is thought to play a role in the inflammatory response. Rubic et al. showed that succinate accumulation activated SUCNR1, increasing inflammatory cytokine production in both human and mouse DCs [18]. Furthermore, Littlewood-Evans et al. observed that inflammatory macrophages of antigen-induced arthritic mice accumulated extracellular succinate, sequentially binding SUCNR1 [21]. Activation of the succinate–SUCNR1 axis resulted in increased IL-1β secretion and, as such, inflammation was reduced in mice deficient for SUCRN1. Although there is strong evidence to suggest the succinate–SUCNR1 axis induces a pro-inflammatory state in macrophages, anti-inflammatory effects of succinate and SUCNR1 have also been elucidated for numerous diseases. Wu et al. observed that cancer cell-derived succinate in the tumor microenvironment can polarize tumor-associated macrophages (TAMs) to a suppressive phenotype through SUCNR1 binding. This was confirmed when silencing of SUCNR1 led to a lack of anti-inflammatory genes *Retnla*, *Arg1*, and *Clec10a* being expressed [22]. In a study focused on succinate’s role in obesity, Keiran et al. showed a similar effect when myeloid-specific SUCNR1 deficiency resulted in upregulation of pro-inflammatory genes *Il1b*, *Il6* and *Il12b* [23]. The anti-inflammatory effect of the succinate–SUCNR1 axis in attribution to obesity and cancer convincingly suggests a role of succinate and SUCNR1 in driving anti-inflammatory responses.

Although in recent years researchers have demonstrated both the pro- and anti-inflammatory effects of the succinate–SUCNR1 axis in macrophages, conclusions are drawn from the use of two types of succinate forms—diethyl succinate, a synthetically generated cell-permeable form; and di-sodium succinate, a cell non-permeable form whose sodium ions dissipate upon dissolving in water, leaving the naturally abundant form present. Additionally, other articles investigate the succinate–SUCNR1 axis with the use of SUCNR1-deficient mice. Therefore, in this paper, we aimed to clarify the immunomodulatory effects of succinate by incorporating all three models into an experimental comparison. Here, we show that succinate can inhibit inflammatory response in macrophages via a SUCNR1-independent manner, while SUCNR1 likely also plays a role in dampening the inflammatory response.

## 2. Results

### 2.1. Cell-Permeable Diethyl Succinate Reduces Secretion and Expression of Inflammatory Mediators in Macrophages

In a previous paper from our lab, we detected a significant induction of succinate in bone marrow-derived macrophages (BMDMs) upon LPS stimulation [24]. Since this confirmed previous observations, we set out to re-evaluate its effect on inflammatory responses in macrophages [16]. Hereto, BMDMs were pre-treated for 1 h with different concentrations of diethyl succinate, followed by 24 h activation with lipopolysaccharide (LPS) or LPS + interferon-gamma (IFN-γ). Diethyl succinate suppressed the LPS (+/-IFN-γ)-induced secretion of the inflammatory mediators interleukin-6 (IL-6), tumor necrosis factor (TNF) and nitric oxide (NO) in a dose-dependent manner (Figure 1A–C). These effects were not due to toxicity since diethyl succinate-treated macrophages were more viable than those left untreated, as evidenced by fixable viability staining in these cells (Appendix A). Since *Il1b* expression was previously shown to be regulated by LPS-induced succinate accumulation [16], we also assessed *Il1b* expression (Figure 1D). In our setting, diethyl succinate did not increase LPS-induced *Il1b* expression as expected and instead decreased it.

While succinate has been well described for its induction of transcription factor HIF-1α protein expression and inflammatory cytokine production, how succinate affects classical macrophage surface biomarker expression has not been defined [16,21,25]. Therefore, using the same culturing conditions as previously stated, we investigated the effect of diethyl succinate on the expression of proteins associated with classical macrophage activation (Figure 1E; gating strategy and representative cytometry data in Appendix A). In line with our above-mentioned observations, we observed that diethyl succinate pre-treatment downregulated CD40 and CD86 surface expression and intracellular iNOS levels in LPS (+/-IFN-γ)-stimulated BMDMs. These findings suggest that in contrast to some previous articles [16,21], diethyl succinate has the ability to inhibit macrophage activation in LPS (+/-IFN-γ)-stimulated cells by suppressing inflammatory cytokine, gene and cell marker expression.

Given that succinate is known to accumulate in LPS- and LPS + IFN-γ-stimulated cells at least partially due to succinate dehydrogenase (SDH) inhibition [15], we simulated this effect by pre-treating BMDMs with dimethyl malonate (DMM) as a competitive inhibitor of SDH [16,25,26]. DMM-mediated inhibition of SDH caused a similar dose-dependent reduction in IL-6, TNF and NO secretion as observed in diethyl succinate-treated macrophages (Figure 2). Together, these data show that increased succinate levels, elicited by exogenous pre-treatment with diethyl succinate or through SDH inhibition with DMM, reduce the expression of pro-inflammatory mediators in macrophages.

### 2.2. The Succinate Receptor SUCNR1 Suppresses Classically Activated Macrophages

Extracellular succinate is reported to induce p38 MAPK signaling through the G protein-coupled receptor 91 (GPR91; i.e., succinate receptor SUCNR1), thereby increasing transcription of at least some pro-inflammatory genes [18,21]. Therefore, we used BMDMs from SUCNR1-KO and control mice in order to investigate the role of SUCNR1 on LPS (+/-IFN-γ)-induced expression of pro-inflammatory mediators. We detected that SUCNR1-deficient macrophages had an increased inflammatory state, identified by increased secretion of IL-6, TNF and NO (Figure 3A–C) upon LPS (+/-IFN-γ) activation, and a trend towards increased *Il1b* expression (Figure 3D) in comparison to their control counterparts. When measuring the expression of membrane-bound markers, CD40 and CD86 were unaffected in LPS (+/-IFN-γ)-stimulated SUCNR1-KO macrophages (Figure 3E). Intracellular levels of iNOS were slightly higher in SUCNR1-KO BMDMs, however, not significantly. Together, these results show that SUCNR1 regulates inflammatory macrophage activation in a negative manner, lowering *Il1b* expression at the mRNA level and IL-6 and TNF secretion at the protein level.

### 2.3. The Anti-Inflammatory Effect of Succinate is Mediated through SUCNR1-Independent Mechanisms

Since we used diethyl succinate in the experiments listed above, the observed effects of this cell-permeable succinate form could be mediated through both intracellular or surface receptor-dependent pathways [9,27]. Therefore, we tested the effect of cell-permeable diethyl succinate on the inflammatory response in WT versus SUCNR1-KO macrophages. Succinate inhibited the LPS (+/-IFN-γ)-induced expression of pro-inflammatory mediators and cell surface markers to a similar extent in WT and SUCNR1-deficient macrophages (Figure 4A–D). To confirm that the effect of succinate acts via a SUCNR1-independent intracellular mechanism, we pre-treated BMDMs with non-cell-permeable disodium succinate, followed by LPS (+/-IFN-γ) stimulation. Succinate did not affect inflammatory responses in macrophages as measured by the secretion of inflammatory mediators and the expression of surface markers (Figure 5A–D). Although succinate has been described to mediate an anti-inflammatory response through SUCNR1 [22,23], our data show that the anti-inflammatory effects are mediated through a SUCNR1-independent mechanism.

## 3. Discussion

In this work, we aimed to clarify how different succinate formulations modulate the inflammatory response of macrophages and how its receptor SUCNR1 partially mediates this. Here, we have shown that cell-permeable diethyl succinate elicits anti-inflammatory responses independently of the succinate receptor SUCNR1. Moreover, despite the fact that exogenous non-permeable succinate did not induce a response through SUCNR1, we observed an anti-inflammatory role for SUCNR1 since KO macrophages had an increased pro-inflammatory profile. Consequently, we have highlighted that succinate, although originally thought to be a pro-inflammatory metabolite, has the potential to induce anti-inflammatory responses in macrophages. What determines whether succinate is utilized in a pro- or anti-inflammatory manner still requires investigation.

Inflammatory macrophages are known to undergo vast metabolic rewiring in order to meet biosynthetic demands, resulting in accumulation of various metabolites [2,4,6,9,15]. Succinate, one of the most well described metabolites to accumulate in these macrophages, has been investigated for its role as an immunomodulatory mediator for inflammation. Initial articles describe succinate as a pro-inflammatory immunometabolite that elicits responses through HIF-1α and subsequent IL-1β secretion in addition to driving production of mitochondrial ROS [16,25]. Moreover, the cell surface receptor SUCNR1 has been shown to also induce IL-1β secretion upon succinate binding [21]. More recent studies demonstrate how succinate and SUCNR1 can also promote anti-inflammatory responses in macrophages and other cell types [22,23,27]. Since these articles use two different forms of succinate, namely a cell-permeable form (diethyl succinate) or the naturally occurring non-permeable form (disodium succinate), in addition to a SUCNR1-deficient mouse model, this asked for a comparison of these three different models in an identical experimental setting.

Succinate dehydrogenase activity is described to be altered in inflammatory macrophages and therefore can cause succinate accumulation [11,16]. Nevertheless, this enzyme still plays a role in succinate oxidation and generation of ROS in these macrophages, which has been shown to drive inflammatory responses [25]. Our data show that treating macrophages with cell-permeable succinate lowers secretion of pro-inflammatory cytokines IL-6, TNF and NO and dampens classical macrophage cell surface marker and *Il1β* mRNA expression. While inhibition of iNOS, and thus NO production, shows how succinate can facilitate reduced macrophage activation, we also observe the capability for succinate to reduce expression of co-stimulatory factors CD40 and CD86 which are expected to stimulate the adaptive immune system. This observation is in apparent contrast to the previously described pro-inflammatory effects of intracellular succinate and therefore we wanted to explore succinate accumulation in an alternative model [16]. Mills et al. observed that the use of DMM to inhibit SDH, an effect which is seen in pro-inflammatory macrophages, additionally caused accumulation of endogenous succinate [25]. Using this model, we aimed to further support the notion that succinate may have the potential to induce anti-inflammatory responses through an intracellular mechanism. Here, we observed that DMM resulted in a dose-dependent inhibition of inflammatory mediators IL-6, TNF and NO, complementing our data on succinate-treated macrophages. This suggests a role for intracellular succinate to inhibit inflammatory responses rather than extracellular succinate. However, whether the intracellular succinate generated due to DMM is transported extracellularly and elicits membrane-mediated responses, or whether further inhibition of SDH and subsequent ROS synthesis is responsible for these observations, requires further studying.

Here, we demonstrate that BMDMs deficient in SUCNR1 secrete higher concentrations of IL-6, TNF and NO as well as having slightly higher levels of classical macrophage activation marker iNOS after stimulation with LPS + IFN-γ. By completely removing the signal using a KO model, macrophages have a reduced ability to suppress pro-inflammatory responses. However, when permeable succinate is added to these SUCNR1-deficient macrophages, succinates’ anti-inflammatory effect is still retained. These observations suggest that while permeable succinate has the capacity to regulate the immune response independent of SUCNR1, signaling through SUCNR1 augments this anti-inflammatory response further. This supports the earlier findings that succinate binding to SUCNR1 induces an anti-inflammatory feedback mechanism in BMDMs [22,23]. Moreover, Kieran et al. described that SUCNR1 is significantly higher expressed in BMDMs stimulated with IL-4 in comparison to LPS, sustaining the idea that the succinate–SUCNR1 axis plays a more important role in mediating a protective phenotype [23]. Indeed, the succinate–SUCNR1 axis was previously described to increase the expression of genes associated with alternative macrophage activation including *Arg1*, *Mrc1*, *Il10* and *Retnla* [23], which is supported by our own observations that arginase activity is increased in succinate as well as DMM-treated macrophages (Appendix A).

In order to better understand the mechanism by which succinate lowers the pro-inflammatory profile of BMDMs, we compared cell-permeable succinate with the non-permeable, naturally occurring form of succinate. As mentioned above, permeable succinate lowered both pro-inflammatory cytokine production and cell surface marker expression; however, the non-permeable form of succinate was unable to elicit these responses. This could be due to the fact that LPS (+/-IFN-γ)-stimulated BMDMs produce high concentrations of succinate and release succinate into the extracellular milieu [11,27,28]. This secretion of succinate may saturate the SUCNR1 signal, therefore masking the binding effect of additional succinate. Alternatively, SUCNR1 could be either desensitized or internalized as has previously been described in other cell types, rendering the succinate–SUCNR1 axis unavailable for further signaling [29,30]. Nonetheless, our data suggest that the anti-inflammatory effect observed by permeable succinate is through an unknown intracellular process. Potentially, succinate could be metabolized to fuel the production of other metabolites with potential anti-inflammatory effects in macrophages. Another intracellular mechanism could be through protein succinylation as this has been shown to adapt protein function, inducing distinct inflammatory roles, such as succinylation of lysine on PKM2 which induced IL-1β secretion [31,32]. However, whether succinylation of proteins that cause repurposing towards driving anti-inflammatory responses in classically activated macrophages occurs, remains to be addressed.

While our observations are indicative of an anti-inflammatory role for both succinate and SUCNR1, we are still striving to explain the discrepancy between our data and other articles that show succinate or SUCNR1 inducing pro-inflammatory responses [16,21,25]. Considering that we used LPS from the same *Escherichia. coli* serotype (O55:B5), followed by identical LPS concentrations and stimulation times as Tannahill et al. or Mills et al., we can only assume that minor differences in culture methods, such as medium (RPMI-1640 vs. DMEM) or fetal calf serum (FCS), results in these differences in response to succinate [16,25]. To further understand what could cause this, we pre-treated macrophages with succinate for 3 h instead of 1 h, as performed in both Tannahil et al. and Mills et al. (data not shown) [16,25]. This resulted in succinate inducing anti-inflammatory responses in macrophages to a similar extent to the 1 h treated macrophages. Although the inhibition of cytokine production and cell marker expression was not as strong as those treated for 1 h, we still observed significance. Nevertheless, we observed a much clearer and significant suppression of *Il1b* mRNA in 3 h of succinate-treated cells compared with the previous 1 h treatment. Therefore, our data demonstrate that the effect of succinate on inflammatory signaling in macrophages is anti-inflammatory, even after attempting to exactly replicate previous articles’ experimental conditions. This leads us to believe that the inflammatory effects of succinate are highly context dependent and should be considered as a dynamic immunometabolite that can induce various inflammatory responses [16,22]. Immunometabolism is a quickly expanding field that could give us further insight into the pathophysiology of many diseases by studying how metabolism can affect inflammation [10,33]. More specifically, this study highlights the role of succinate in regulating inflammatory responses which can later be applied to diseases that display altered local or systemic succinate levels. In particular, the synovium fluid of patients with the chronic autoimmune disease rheumatoid arthritis has been shown to have increased succinate levels [21]. By observing the immunomodulatory properties of succinate as well as its derivatives, as done so in this study, we can better consider how to develop appropriate treatments for such diseases. Since it has also been shown that succinate is one of the most highly induced metabolites in pro-inflammatory macrophages, applying the knowledge gained from studying succinate should stretch to inflammatory diseases beyond that of rheumatoid arthritis [24].

Together, these data show that succinate can induce anti-inflammatory responses in BMDMs, most likely intracellularly, in addition to SUCNR1 further augmenting this effect. However, since this is in contradiction to many initial articles describing the immunomodulatory effects of succinate, this generates a lot of unanswered questions on how succinate is truly regulating inflammation in macrophages and requires an appropriate human setting to explore this. These data, in combination with those of other research institutes, highlight the complexity of immunometabolism and that succinate and SUCNR1 may play a more diverse role than originally anticipated.

## 4. Materials and Methods

### 4.1. Mice

Wild-type C57BL/6J(c) 8- to 16-week-old mice were obtained from Charles River Laboratories, and bone marrow of 40-week-old Sucnr1^−/−^ mice and WT controls, generated as previously described [34], was generously gifted from Peter Deen (Radboud University Nijmegen). All experiments were approved by the Committee for Animal Welfare (Vrije Universiteit Amsterdam, Amsterdam, The Netherlands).

### 4.2. Macrophage Cultures

Mouse bone marrow cells were isolated and cultured in RPMI-1640 with 2 mM L-glutamine, 10% fetal calf serum (FCS), penicillin (100 U/mL), streptomycin (100 μg/mL) (Gibco), and 15% L929-conditioned medium. On day 6, cells were harvested, seeded at 10^6^ cells/mL and treated for 1 h with succinate or DMM at a concentration of 5 or 10 mM, respectively, unless otherwise stated. Following succinate or DMM treatment, BMDMs were stimulated for 4 or 24 h with 100 ng/mL LPS serotype O55:B5 (Sigma-Aldrich), 10 U/mL IFN-γ (PeproTech) + 10 ng/mL LPS or 20 U/mL IL-4 (Peprotech) to generate pro-inflammatory (LPS and LPS + IFN-γ) or anti-inflammatory (IL-4) macrophages, respectively. After stimulation, the supernatant was harvested for use in functional assays and macrophages were washed with PBA and harvested for phenotyping by flow cytometry and gene expression analysis.

### 4.3. Macrophage Function

IL-6 and TNF were quantified by ELISA in accordance with the supplier’s protocols (Life Technologies). NO production was measured by NO_2_^−^ quantification in a Griess reaction (Sigma-Aldrich). Arginase activity (1 U = amount of enzyme that catalyzes the formation of 1 μmol urea/min/10^6^ cells) was assessed as described previously [35].

### 4.4. Gene Expression Analysis

RNA was isolated with GeneJET RNA Purification kits (ThermoFisher), cDNA was synthesized with a High-Capacity cDNA Reverse Transcription Kit (Applied Biosystems) and quantified using an Implen nanodrop N60. qPCR was performed using a Sybr Green Fast mix (Applied Biosytems) on a ViiA7 (Applied Biosystems). Housekeeping genes *Rplp0* (*Arbp*) and *Cycloa* were used for normalization. Primer sequences are available upon request.

### 4.5. Flow Cytometry

Cells were incubated with Fc block in addition to labeling antibodies and dyes listed in Appendix A for 1 h. Cell viability was assessed by fixable viability staining according to the manufacturer’s protocol (eBioscience). Data were acquired with an LSRFortessa X20 (Becton Dickinson) and analyzed using FlowJo v10 (Becton Dickinson) or FCSExpress v7 (De Novo Software). Surface expression was calculated as ΔMFI = (median fluorescence intensity) positive staining—(median fluorescence intensity) negative staining/background fluorescence.

A full list of antibodies, dyes and chemicals used in this article can be found listed in Appendix A.

### 4.6. Statistical Analysis

Data are presented as the mean ± standard deviation (SD) and were tested using either one- or two-way ANOVA with a Bonferroni post hoc test in GraphPad Prism version 8.2.1 software. *p* values < 0.05 were considered significant, with levels of significance being indicated as follows: * *p* < 0.05; ** *p* < 0.01; *** *p* < 0.001; **** *p* ≤ 0.0001.

## Figures and Tables

**Figure 1 metabolites-10-00372-f001:**
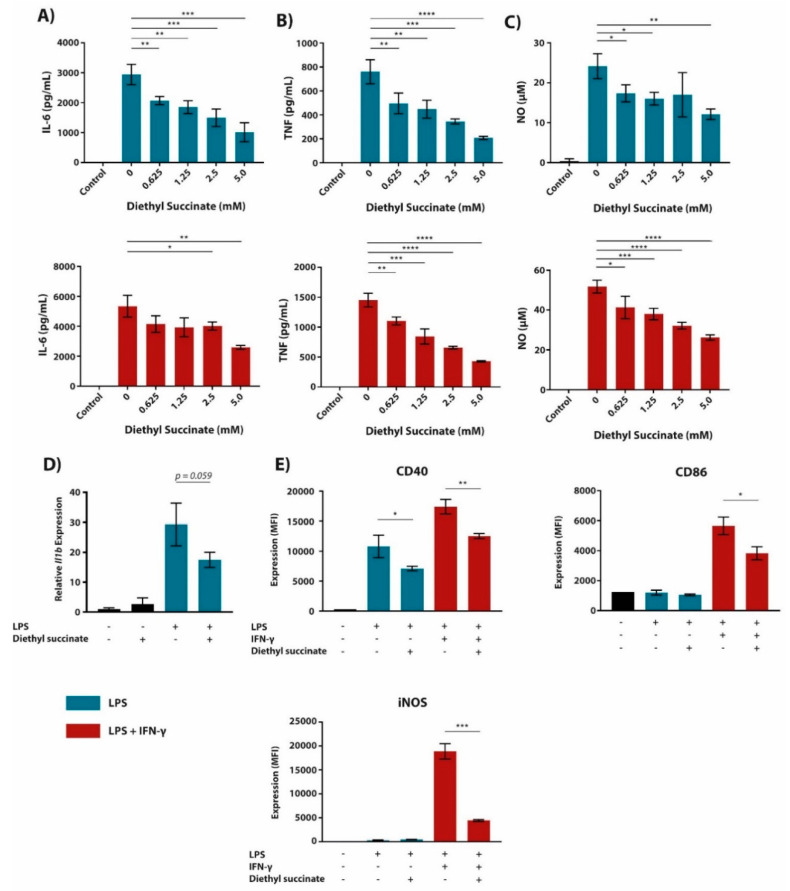
Succinate reduces pro-inflammatory cytokine secretion, gene and marker expression in BMDMs. (**A**–**E**) BMDMs were pre-treated with diethyl succinate (0.625–5.0 mM; **A**–**C** or 5 mM; **D**,**E**) 1 h prior to LPS (100 ng/mL; blue bars) or LPS + IFN-γ (10 ng/mL + 10 U/mL, respectively; red bars) stimulation for 24 h (**A**–**C**,**E**) or 4 h (**D**). The supernatant was then analyzed for IL-6 (**A**) and TNF (**B**) cytokine secretion by ELISA. Additionally, NO secretion (**C**) was measured by use of Griess Reagent. Subsequently, cells were either lysed and mRNA extracted from cell lysate to measure gene expression of *Il1b* by qPCR (**D**) or stained with fluorescently labelled antibodies against CD40, CD86 and iNOS (**E**). Stained cells were then read by a BD LSRFortessa X-20 and cell biomarker surface expression was analyzed in FCSExpress 7. All data are in triplicate (**A**–**D**) or quadruplicates (**E**) represented as the mean ± SD and are representative of three independent experiments. Controls are unstimulated BMDMs. * *p* ≤ 0.05, ** *p* ≤ 0.01, *** *p* ≤ 0.001, and **** *p* ≤ 0.0001.

**Figure 2 metabolites-10-00372-f002:**
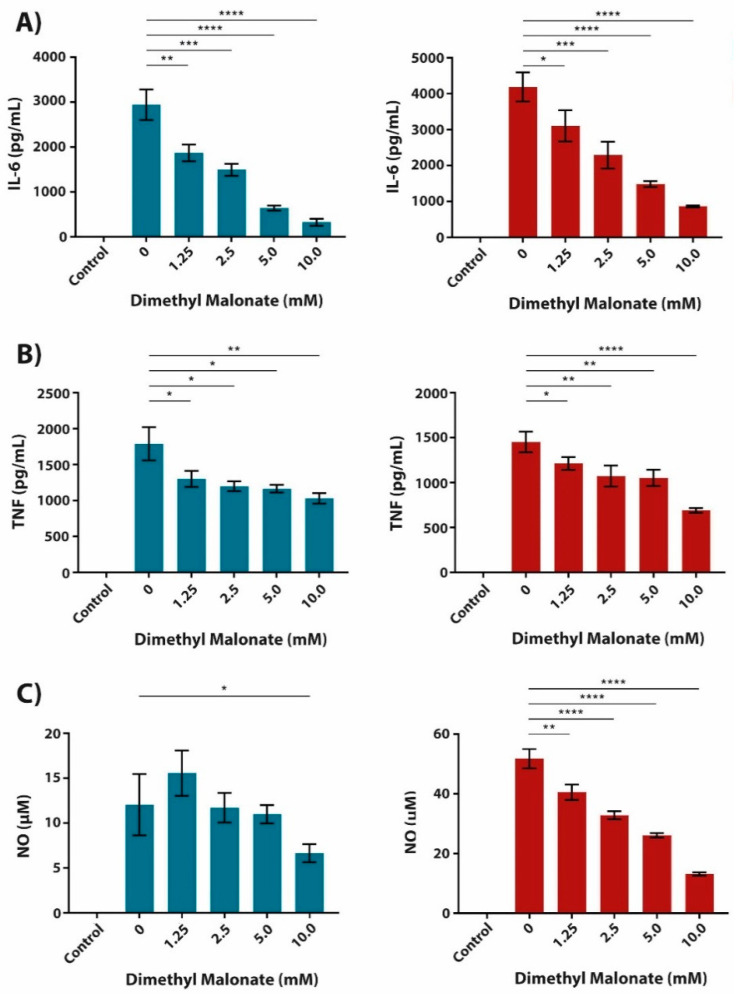
Cell-permeable malonate, an inhibitor of SDH, elicits similar functional responses in pro-inflammatory BMDMs as cell-permeable succinate. (**A**–**C**) BMDMs were pre-treated with dimethyl malonate (DMM) 1 h prior to LPS (100 ng/mL; blue bars) or LPS + IFN-γ (10 ng/mL + 10 U/mL, respectively; red bars) stimulation for 24 h. The supernatant was then analyzed for IL-6 (**A**) and TNF (**B**) cytokine secretion by ELISA. Additionally, NO secretion (**C**) was measured by use of Griess Reagent. All data are in quadruplicates represented as the mean ± SD and are representative of three independent experiments. * *p* ≤ 0.05, ** *p* ≤ 0.01, *** *p* ≤ 0.001, and **** *p* ≤ 0.0001.

**Figure 3 metabolites-10-00372-f003:**
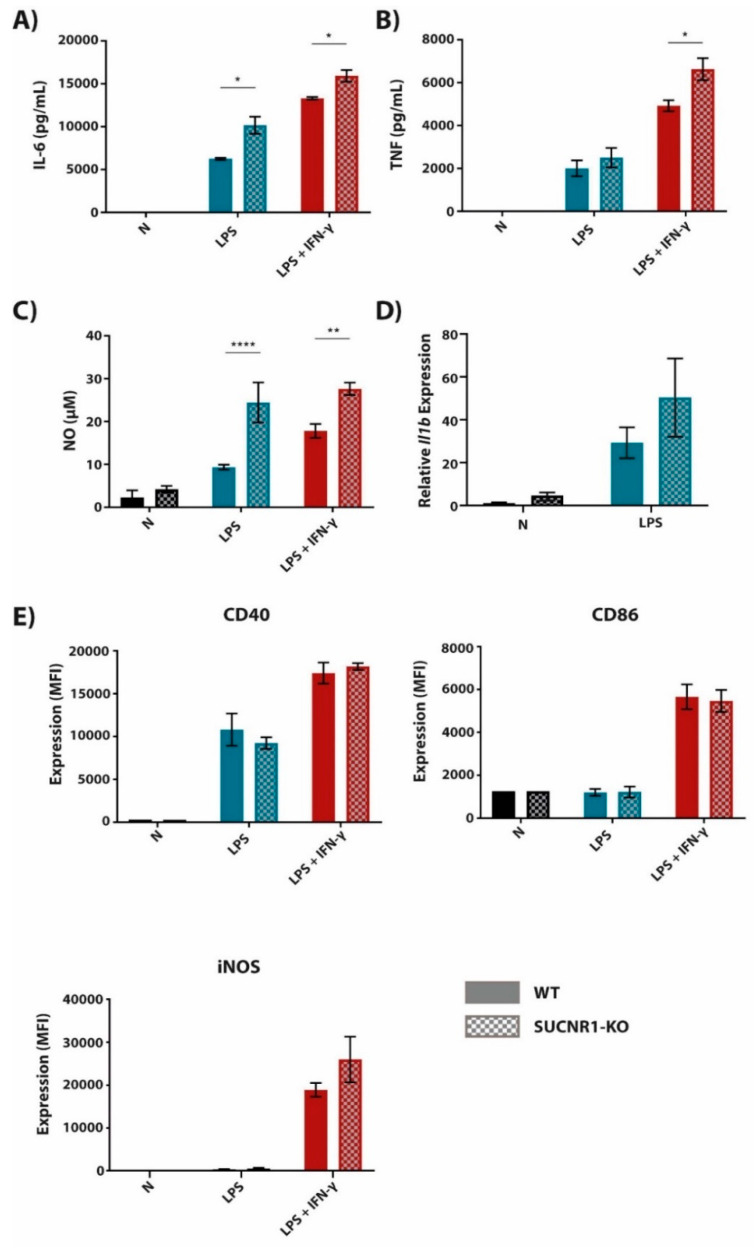
BMDMs deficient in succinate receptor SUCNR1 have an increased pro-inflammatory phenotype. BMDMs derived from SUCNR1-KO and WT littermate controls were stimulated with LPS (100 ng/mL) or LPS + IFN-γ (10 ng/mL + 10 U/mL, respectively) for 24 h (**A**–**C**,**E**) or 4 h (**D**). The supernatant was then analyzed for IL-6 (**A**) and TNF (**B**) cytokine secretion by ELISA. Additionally, NO secretion (**C**) was measured by use of Griess Reagent. Subsequently, cells were lysed and mRNA was extracted from cell lysate to measure gene expression of *Il1b* by qPCR (**D**). Cells were also stained with fluorescently labelled antibodies against CD40, iNOS and CD86 (**E**). Stained cells were then read by a BD LSRFortessa X-20 and cell biomarker surface expression was analyzed in FCSExpress 7. All data are in quadruplicates represented as the mean ± SD and are representative of three independent experiments. * *p* ≤ 0.05, ** *p* ≤ 0.01 and **** *p* ≤ 0.0001.

**Figure 4 metabolites-10-00372-f004:**
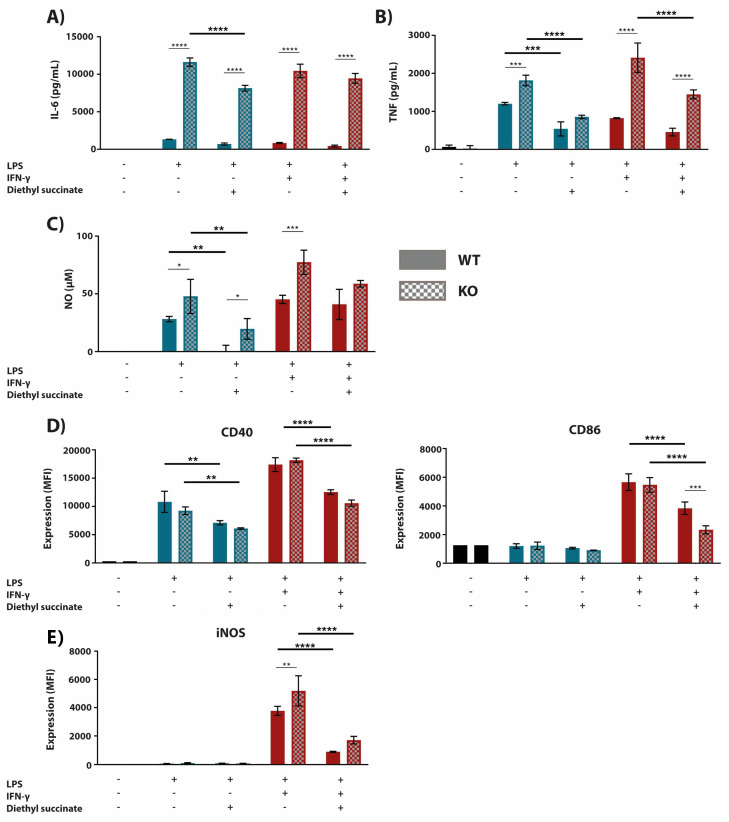
BMDMs deficient in succinate receptor SUCNR1 express a dampened pro-inflammatory phenotype in response to succinate. BMDMs derived from SUCNR1-KO and WT littermate controls were pre-treated with succinate (5 mM) 1 h prior to stimulation with LPS (100 ng/mL) or LPS + IFN-γ (10 ng/mL + 10 U/mL, respectively) for 24 h (**A**–**C**,**E**) or 4 h (**D**). The supernatant was then analyzed for IL-6 (**A**) and TNF (**B**) cytokine secretion by ELISA. Additionally, NO secretion (**C**) was measured by use of Griess Reagent. Cells were subsequently stained with fluorescently labelled antibodies against CD40, iNOS and CD86 (**D**). Stained cells were then read by a BD LSRFortessa X-20 and cell biomarker surface expression was analyzed in FCSExpress 7. All data are in quadruplicates represented as the mean ± SD and are representative of three independent experiments. Statistical significance indicated in bold and larger asterisks are used to highlight essential comparisons of this figure, whereas smaller asterisks show significant comparisons which have already been made apparent in previous figures. * *p* ≤ 0.05, ** *p* ≤ 0.01, *** *p* ≤ 0.001, and **** *p* ≤ 0.0001.

**Figure 5 metabolites-10-00372-f005:**
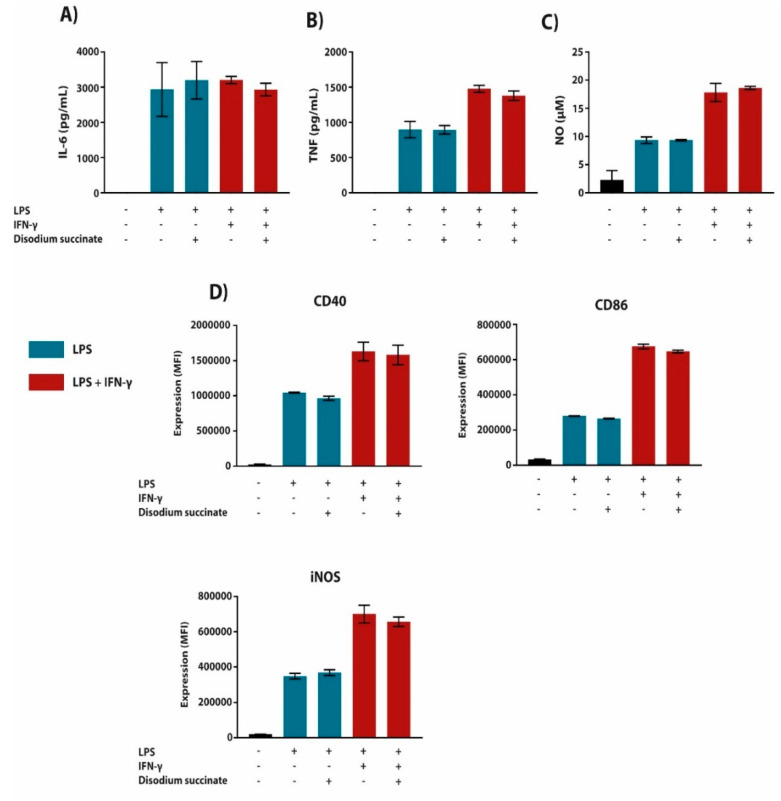
Cell non-permeable succinate has no effect on cytokine secretion or cell surface biomarker expression in pro-inflammatory BMDMs. (**A**–**D**) BMDMs were pre-treated with disodium succinate (5 mM) 1 h prior to LPS (100 ng/mL) or LPS + IFN-γ (10 ng/mL + 10 U/mL, respectively) stimulation for 24 h. The supernatant was then analyzed for IL-6 (**A**) and TNF (**B**) cytokine secretion by ELISA. Additionally, NO secretion (**C**) was measured by use of Griess Reagent. Cells were subsequently stained with fluorescently labelled antibodies against CD40, iNOS and CD86 (**D**). Stained cells were then read by a BD LSRFortessa X-20 and cell biomarker surface expression was analyzed in FCSExpress 7. All data are in quadruplicates represented as the mean ± SD and are representative of three independent experiments.

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
