# Peer review of "Succinate Is an Inflammation-Induced Immunoregulatory Metabolite in Macrophages"

_metabolites, 2020, doi:10.3390/metabo10090372_

Round 1

Reviewer 1 Report

Karl Harber et al present nice studies aiming to investigate the role of succinate in macrophages. There is some conflicting data in the literature and the authors wanted to compare all approaches in the literature present. This is an important study comparing three different methods to modulate succinate metabolism in macrophages.

I only have some minor concerns:

The pro- versus anti-inflammatory effects that were and are described might be a time-dependent effects. Does succinate increased IL-1b mRNA expression after three hours as described in the early reports? How is IL-1B protein expression modulated after 24h with LPS+ATP?

Page 97: The sentence “Having confirmed…” is not clearly formulated, the reader would expect the results in the present manuscript, not in an earlier paper.

Page 164: The first sentence should be referenced.

The discussion might also include a statement a about culture conditions. I could not find detailed descriptions of the culture conditions of BMDMs in the Nature 2013 paper or the referenced method paper.

Reviewer 2 Report

The aim of this article is to clarify the immunomodulatory effect of succinate.  Although the use of different forms of succinate to identify their effect on macrophages is important, it is not provided a potential mechanism to explain the differential inflammatory effect of this metabolite.  While the experiments presented in this manuscript are well designed, there are several concerns in the presentation and the interpretation of the results.   Following are some specific concerns about the manuscript:

Major concerns:

  1. Figure 1. It is not indicated the difference of green and red graphics in A-C.  Image of FACS should be provided.
  2. As in figure 1, images of FACS should be included.
  3. Line 138, it is concluded SUCNR1 decreases cytokine secretion at both protein and mRNA level. However, IL-1β is the only cytokine detected at transcriptional level.  It should be clarified this cytokine was reduced at mRNA level and IL-6 and TNFα at protein level.
  4. Line 244. Since this is a small manuscript data about the increase in the activity of arginase in succinate treated macrophages should be included.
  5. The authors at least should include a potential hypothesis to explain the different effect of succinate in addition to the protein succinylation that has been previously proposed.

Minor concern:

  1. Line 163-164. The title of this section (line 163) and the first paragraph (164) are the same.  Line 164 should be removed.

Reviewer 3 Report

This is an interesting study discussing the area of metabolism on macrophage function/activation.  The content appears scientifically sound though the cohesion of the experimental design is hard to follow.

I am still unsure what the authors mean by "context dependent", could they clarify for the reader.

I was particularly interested in the data on malonate compared to succinate.  This should be in the main body of the paper and adequately discussed.

The discussion does not adequately discuss the potential biomedical significance of the findings.  I am left with the opinion of "so what" when considering the findings to inflammatory disease.  Could the authors provide their insight on what they consider the significance.  Including a graphical summary in discussion of the findings would be helpful.  This is where consideration of the similarities of succinate and malonate structures would be helpful.  Also l am unclear if there is any physiological significance of cell impermeable compared to cell permeable succinate.

The choice of markers for study could be elaborated on.  There is "negative data" that might be significant.  I am thinking about CD40 and CD86.  Also Fig 3C, NO.  I am surprised that the effect of lPS/IFN in the KO was not significant.  Can the authors verify.

Can the authors explain how their viability assay compares to the more traditional MTT/WST mitochondrial activity assays.  Do they have comparable data.  The stimulation effect is hard to explain for the assay used.  More details on that methodology are required.

Round 2

Reviewer 2 Report

This is a revised version of the paper entitled “Succinate is an inflammation-induced immunoregulatory metabolite in macrophages”.  The authors have undertaken further work, which significantly extends the scientific interest of this manuscript.  Most of the points raised in the previous review have been addressed.